# A diagnostic primer pair to distinguish between *w*Mel and *w*AlbB *Wolbachia* infections

**Meng-Jia Lau** **\*, Ary A. Hoffmann, Nancy M. Endersby-Harshman**

Pest and Environmental Adaptation Research Group, Bio21 Institute and the School of BioSciences, The University of Melbourne, Parkville, Victoria, Australia

\* mengjial2@student.unimelb.edu.au

## Abstract

Detection of the *Wolbachia* endosymbiont in *Aedes aegypti* mosquitoes through real-time polymerase chain reaction assays is widely used during and after *Wolbachia* releases in dengue reduction trials involving the *w*Mel and *w*AlbB strains. Although several different primer pairs have been applied in current successful *Wolbachia* releases, they cannot be used in a single assay to distinguish between these strains. Here, we developed a new diagnostic primer pair, *wMwA*, which can detect the *w*Mel or *w*AlbB infection in the same assay. We also tested current *Wolbachia* primers and show that there is variation in their performance when they are used to assess the relative density of *Wolbachia*. The new *wMwA* primers provide an accurate and efficient estimate of the presence and density of both *Wolbachia* infections, with practical implications for *Wolbachia* estimates in field collected *Ae. aegypti* where *Wolbachia* releases have taken place.

## Introduction

The bacterium, *Wolbachia*, is providing an increasingly popular method to inhibit dengue virus transmission in the mosquito, *Aedes aegypti*. *Wolbachia*-infected populations involving the *w*Mel strain have now been successfully established in *Ae. aegypti* in regions including northern Australia, Brazil and Indonesia [1–3], while *w*AlbB-infected *Ae. aegypti* have been established in Malaysia [4]. Detection of the *Wolbachia* endosymbiont in *Ae. aegypti* mosquitoes is a standard requirement for good laboratory practice during *Wolbachia* mosquito releases in dengue reduction programs and for tracking *Wolbachia* invasions in the field [4, 5]. Real-time polymerase chain reaction (real-time PCR) and High Resolution Melt (HRM) assays (SYBR® equivalent/non-probe) have been developed that enable detection and *Wolbachia* density estimation for the strain of interest [6–8]. However, difficulties can arise in using these assays when there is a need to detect *Wolbachia* and distinguish between multiple *Wolbachia* strains. In experiments where superinfected lines are used [9], or where mosquitoes carrying different single infections need to be distinguished for experiments or in field collected samples [10], several real-time PCR assays using different primer pairs are currently required. Given that both *w*Mel and

**Data Availability Statement:** All relevant data are within the manuscript and its Supporting Information files.

**Funding:** Ary A. Hoffmann is funded by the National Health and Medical Research Council

(1132412, 1118640, www.nhmrc.gov.au). The funders had no role in study design, data collection and analysis, decision to publish, or preparation of the manuscript.

**Competing interests:** The authors have declared that no competing interests exist.

wAlbB strains are now actively being used in field releases and that each strain may have advantages in particular situations, the requirement for multiple strain identification is likely to increase in the foreseeable future.

In previous work, we have used a *Wolbachia*-specific primer pair, *w1* [7], which targets a conserved locus VNTR-141 containing tandem repeats [11]. This pair of primers works efficiently in amplifying *w*Mel and *w*MelPop infections in a real-time PCR and HRM assay, but achieves poor amplification of *w*AlbB [10]. As well as being used for *Wolbachia* detection, primers are needed for quantification of *Wolbachia* density in mosquitoes. There are various *Wolbachia* specific primers for *w*Mel, *w*AlbB or *w*MelPop [9, 12–14], but currently there is no standardized assay for *Wolbachia* screening that is comparable between strains and that can be used to compare results between laboratories. Although cross-laboratory comparability may not be a realistic aim when using a SYBR® equivalent/non-probe-based assay, the use of extra internal controls can make these assays robust for relative density estimates, improving consistency within laboratory experiments [7, 10].

In this study, we developed a diagnostic primer pair that can detect and distinguish between the *w*Mel and *w*AlbB infections and also provides an estimate of *Wolbachia* density. In addition, we assessed primer efficiency of some other published primers for *Wolbachia* in mosquitoes. We also tested quantification cycle ($C_q$) [15] value differences between primers for different *Wolbachia* strains to assess primer suitability for relative *Wolbachia* density estimation.

## Materials and methods

### Diagnostic primer design

To develop the new primers, we screened for sequence differences between the *w*Mel and *w*AlbB strains and then focused on the sequences of a DNA-directed RNA polymerase subunit beta/betagene with locus tag WD_RS06155 in *w*Mel and its analogue in *w*AlbB. We then developed a new pair of primers designated *wMwA* (Table 1) to distinguish *Wolbachia w*Mel and *w*AlbB in a single run of a real-time PCR assay, based on two base-pair mismatches at the 3'-end of each primer, which resulted in the Tm peak for *w*AlbB being separated from that of *w*Mel. We checked the specificity of this primer pair by an initial test of six males and six females for each strain with different *Wolbachia* infection type (*w*Mel- or *w*AlbB-infected or uninfected). Subsequent testing was done with female mosquitoes only.

### Sample preparation

The *w*Mel and *w*AlbB-infected *Ae. aegypti* were tested for strains transinfected previously [16, 17]. The *w*Mel strain was collected from Cairns, Australia in 2019 from regions that had been invaded several years earlier [2, 12], while the *w*AlbB strain was derived from a *w*AlbB infected strain crossed to an Australian background and maintained in the laboratory [13]. An uninfected strain was developed from *Ae. aegypti* eggs collected in Cairns, Queensland, Australia prior to *Wolbachia* releases [10, 18].

Female mosquitoes of *w*Mel-infected [17], *w*AlbB-infected [16] and uninfected were reared with TetraMin® fish food tablets in reverse osmosis (RO) water until the adult stage [19], and then were killed in absolute ethanol before Chelex® DNA extraction. In the standard procedure, DNA of an individual female was extracted in 250 μL 5% Chelex® 100 Resin (Bio-Rad Laboratories, Hercules, CA) and 3 μL of Proteinase K (20 mg/ mL, Bioline Australia Pty Ltd, Alexandria NSW, Australia). The Chelex® 100 Resin solution containing DNA was centrifuged at 12500 rpm for 5 min and DNA solution was pipetted from the supernatant.

**Table 1. Primers for detection of *Wolbachia* strains and estimation of density.**

| Primer name | *Wolbachia* status target | Targeted locus | Forward | Reverse | Amplicon size (bp) | Source |
|---|---|---|---|---|---|---|
| *mos* | uninfected | AF154067 | AGTTGAACGTATCGTTTCCCGCTAC | GAAGTGACGCAGCTTGTGGTCGTCC | 77 | [7] |
| *aeg* | uninfected | AF154067 | ATCAAGAAGCGCCGTGTCG | CAGGTGCAGGATCTTCATGTATTCG | 66 | [7] |
| *w1* | *w*Mel | VNTR 141 region | AAAATCTTTGTGAAGAGGTGATCTGC | GCACTGGGATGACAGGAAAAGG | 16 | [7] |
| *wsp* | *w*Mel, *w*AlbB-infected | WD_RS04815 | GCATTTGGTTAYAAAATGGACGA | GGAGTGATAGGCATATCTTCAAT | 139 (*w*Mel), 136 (*w*AlbB) | [9] |
| *wMel* | *w*Mel-infected | WD_RS02275 | CAAATTGCTCTTGTCCTGTGG | GGGTGTTAAGCAGAGTTACGG | 68 | [9] |
| *wAlbB* | *w*AlbB-infected | DEJ70_RS01110 | CCTTACCTCCTGCACAACAA | GGATTGTCCAGTGGCCTTA | 109 | [9] |
| *wMwA* | *w*Mel, *w*AlbB-infected | WD_RS06155 | GAAGTTGAAGCACAGTGTACCTT | GCTTGATATTCCTGTAGATTCATC | 155 (both) | Newly designed |

## LightCycler® efficiency test

After extraction, DNA concentration was measured using a Qubit™ 1X dsDNA HS Assay Kit and Qubit™ 2.0 fluorometer (ThermoFisher Scientific, Waltham, MA USA), and then diluted ten times before making a three-fold dilution series to test the efficiency of currently-used *Wolbachia* primers in a real-time PCR assay (Table 1). We also diluted the solution six times before making a three-fold dilution series to investigate the influence of Chelex®-extracted DNA concentration.

For the real-time PCR and HRM, we used a LightCycler® 480 High Resolution Melting Master (HRMM) kit (Roche; Cat. No. 04909631001, Roche Diagnostics Australia Pty. Ltd., Castle Hill New South Wales, Australia) and IMMOLASE™ DNA polymerase (5 U/µl) (Bioline; Cat. No. BIO-21047) as described by Lee et al. (2012) (S1 Table). We used 384-well plates with white wells (SSI Bio, Lodi CA USA, Cat. No. 3430–40), and the PCR conditions for DNA amplification beginning with a 10-minute pre-incubation at 95˚C (Ramp Rate = 4.8˚C/s), followed by 40 cycles of 95˚C for 5 seconds (Ramp Rate = 4.8˚C/s), 53˚C for 15 seconds (Ramp Rate = 2.5˚C/s), and 72˚C for 30 seconds (Ramp Rate = 4.8˚C/s).

Three technical replicates were run for each sample of each dilution and a graph was produced showing the log3 [dilution factor] (x-axis) against mean $C_q$ (y-axis) and a linear trend line (y = mx + c) was fitted. Slope (m) and $R^2$ values were recorded so that PCR amplification efficiency (E) could be evaluated with the equation:

$$E = (3^{-\frac{1}{slope}} - 1) \times 100\%$$

Compare with Chelex® extraction, we also purified DNA from the above Chelex® 100 Resin solution using the PureLink™ Quick PCR purification Kit (Invitrogen Cat. No. K3100-01), in which the binding buffer B2 was used. In addition, a different DNA extraction method was used: female mosquitoes were homogenized individually in 100 µL STE buffer (10 mM Tris-HCl pH8, 100 mM NaCl, 1mM EDTA), and then incubated at 95˚C for 10 minutes. After these extractions, 10 µL supernatant was pipetted into 90 µL ddH$_2$O and made a three-fold dilution series.

## Primer quantification cycle comparison and density estimation

Following the efficiency study, we used a mixture of young (4±1days since eclosion) and old (38 ±1days since eclosion) female mosquitoes and tested for $C_q$ value differences between primers for different *Wolbachia* strains to assess suitability for relative *Wolbachia* density

estimation. A total of 16 *Wolbachia*-infected mosquito samples were extracted using Chelex®️ resin and then diluted ten times before real-time PCR.

# Results and discussion

## Diagnostic primer design

In this study, we developed a diagnostic primer pair, *wMwA*, that can detect and distinguish between the *w*Mel and *w*AlbB infections in *Aedes aegypti* (Fig 1), which is important in simplifying current approaches for *Wolbachia* identification. In the initial test for the specificity of this primer pair, all uninfected samples were negative, and all *Wolbachia-*

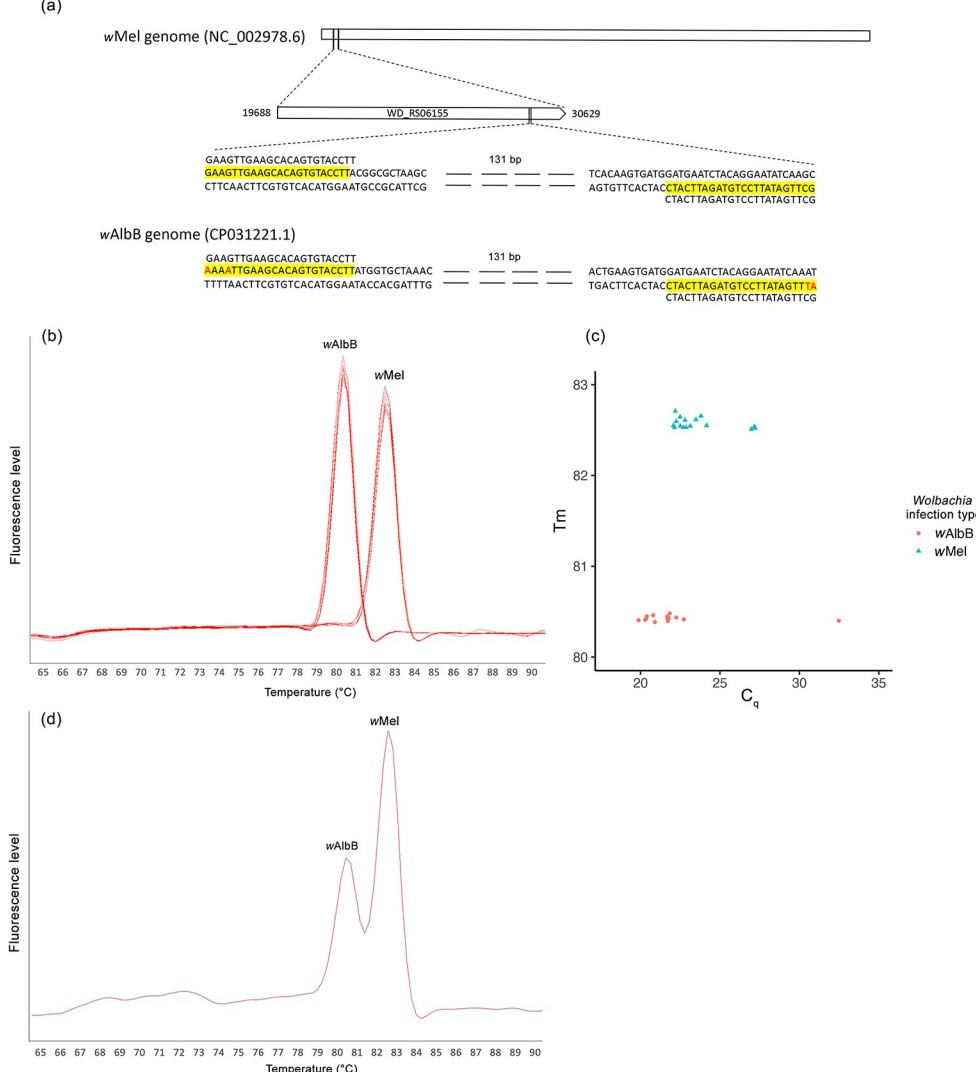

**Fig 1. Development of primers to detect *Wolbachia* *w*Mel and *w*AlbB infection in *Aedes aegypti*.** (a) The new primer pair *wMwA* aligns to a region in gene WD_RS06155 of *w*Mel, and also aligns to its analogue in the *w*AlbB genome which has two base-pair mismatches at the 3'- end; (b) the *wMwA* primers showed distinct Tm peaks for *Wolbachia* *w*Mel and *w*AlbB. (c) the *wMwA* primers showed distinct Tm values for *Wolbachia* *w*Mel (82.6 ± 0.03°C) and *w*AlbB (80.4 ± 0.02°C), the x axis represents the quantification cycle ($C_q$) and the y axis represents the amplicon melting temperature; (d) the *wMwA* primers showed two Tm peaks when mixing DNA templates of *w*Mel and *w*AlbB-infected *Ae. aegypti*.

infected samples were positive with distinctive Tm values from *Wolbachia w*Mel (82.6 ± 0.03˚C) and *w*AlbB (80.4 ± 0.02˚C) screening (Fig 1C). The high-resolution melt produces two joined peaks when the template contains both *Wolbachia w*Mel and *w*AlbB DNA (Fig 1D).

### Primer efficiency test

We tested the efficiency of each of the primers for screening *Wolbachia* in *Ae. aegypti* by using a threefold dilution series. When template DNA was extracted in Chelex® 100 Resin solution, the efficiencies of all primers ranged from 86.4% to 104.9%, (Table 2 and Fig 2) and the efficiency curves all showed an $R^2$ valued greater than 0.99.

However, we found the amplification curve increase showed inhibition at the first dilution (Fig 3) for each of the primers, particularly when DNA was first diluted six times instead of ten times, resulting in outliers (S1 and S2 Figs and S2 Table). These results highlight a potential risk of lowering the relative density estimate in *Wolbachia* screening when using a highly concentrated Chelex®-extracted DNA solution. We also found differences between primer efficiency when a different DNA extraction method was used, with changes ranging from -22.2% to 29% (S3 Table). Different DNA extraction methods may affect DNA yield and quality, and/or change PCR inhibitors and their effects, which can increase variation between host and parasite DNA [20–22]. It is, therefore, worth noting that new standard curves should be run when changing to a different DNA extraction method, given that the efficiency of primers can deviate substantially from recommendations (90% - 110%) [23, 24] to prevent an inaccurate estimate of relative density being made.

### Cq value comparisons in Chelex® 100 Resin

We noticed that primers had different $C_q$ values even when screening the same individual organism/endosymbiont (*Ae. aegypti, w*Mel or *w*AlbB) and using the same DNA concentration, despite the efficiency of these primers all falling within 85% - 110%. We therefore tested the $C_q$ ranges of the primers and correlated them with *wsp*. We found variation between these primers (Fig 2), which would be expected to result in differences in relative density estimates. The relationship between $C_q$ values of different primers all fit into a linear relationship, with $R^2$ greater than 0.97, whereas the coefficient varies from 0.83 to 1.05 (Fig 4). For the newly-designed primer pair *wMwA*, the coefficients for *w*Mel and *w*AlbB are similar (0.97 for *w*Mel and 1.04 for *w*AlbB).

**Table 2. Primer efficiency for primer pairs used in detection of *Wolbachia* strains and estimation of density.**

| Colony | Primers | Slope of graph | $R^2$ | Efficiency | DNA concentration* (ng/μL) | Efficiency curve |
|---|---|---|---|---|---|---|
| Uninfected | *mos* | -1.566 | 0.999 | 101.659% | 5.12 | Fig 2A |
| Uninfected | *aeg* | -1.531 | 0.999 | 104.946% | 5.12 | Fig 2B |
| *w*Mel | *w1* | -1.644 | 0.999 | 95.109% | 5.24 | Fig 2C |
| *w*Mel | *wM* | -1.677 | 0.998 | 92.559% | 5.24 | Fig 2D |
| *w*Mel | *wsp* | -1.767 | 0.999 | 86.195% | 5.24 | Fig 2E |
| *w*Mel | *wMwA* | -1.769 | 0.999 | 86.107% | 5.24 | Fig 2F |
| *w*AlbB | *wA* | -1.741 | 0.993 | 87.953% | 6.87 | Fig 2G |
| *w*AlbB | *wsp* | -1.755 | 0.999 | 87.032% | 6.87 | Fig 2H |
| *w*AlbB | *wMwA* | -1.730 | 0.997 | 88.732% | 6.87 | Fig 2I |

*Template DNA was extracted in 250 μL 5% Chelex® 100 Resin and then diluted ten times before making a three-fold dilution series. Concentration was measured before dilution.

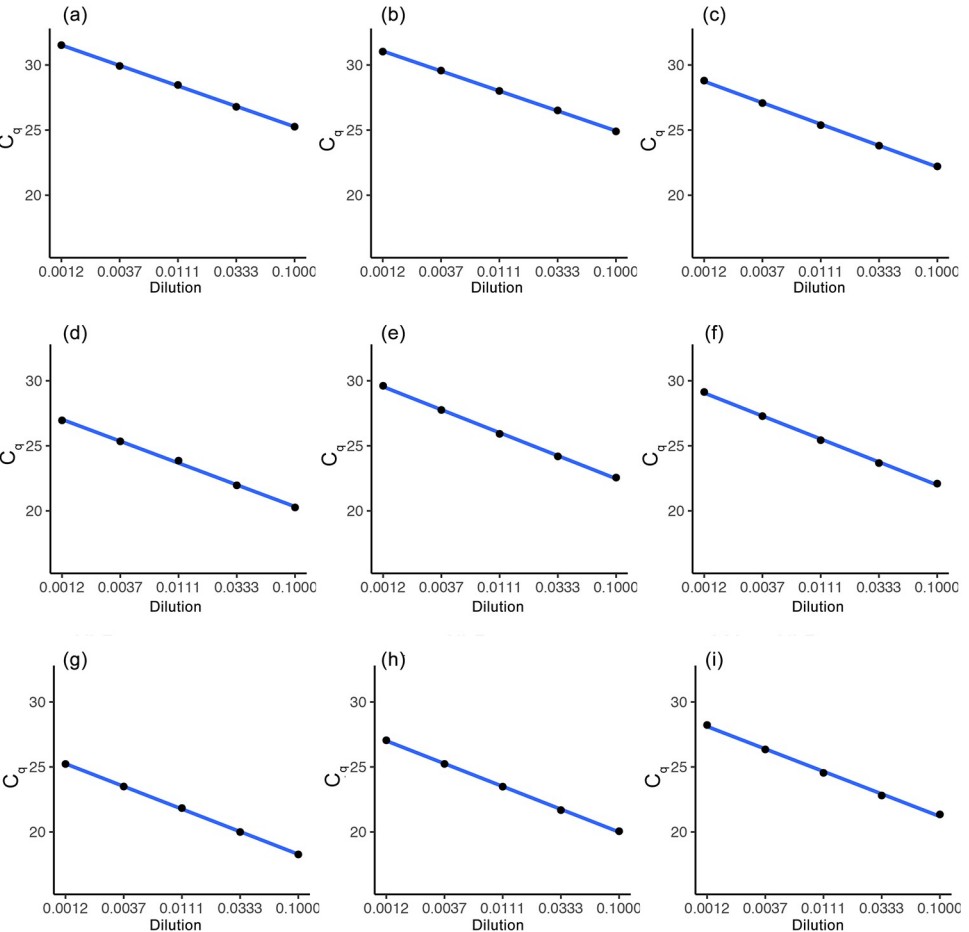

**Fig 2. Primer efficiency for detection of *Wolbachia* strains and estimation of density.** DNA was extracted in 250 μL 5% Chelex® 100 Resin and then diluted ten times before making a three-fold dilution series. The primer names are defined in Table 2.

These primer differences could not be explained fully by pipetting error and PCR inhibition [25, 26]. Inhibition effects on DNA amplification can vary when using different primers, and/or when the DNA concentration varies. Intercepts of these $C_q$ values ranged from -1.52 to 1.37 though all primers used in this study only have one copy based on their genomic sequences. However, it is possible that there may be different copies of *Wolbachia* genes inside mosquito cells [27, 28], such as is documented for the octomom region [29, 30] which can be variable under different environmental conditions [31, 32]. As a result, care is needed when choosing primers for assessing the relative concentration of *Wolbachia*.

In our study, the *wsp* primers represent a useful pair of universal primers for amplifying the *Wolbachia* surface protein gene which has been applied as a *Wolbachia* diagnostic for decades [14]. Given potential variation between *Wolbachia* primers, comparisons with universal *Wolbachia* primers should be undertaken before using the newly-designed primers in *Wolbachia* density calculations. Our newly-designed primer pair, *wMwA*, correlated with density estimates based on *wsp*, with coefficients for both *w*Mel and *w*AlbB close to 1. Thus, this new primer pair has the potential to be accurate and efficient for large-scale *Wolbachia* detection and relatively density estimate.

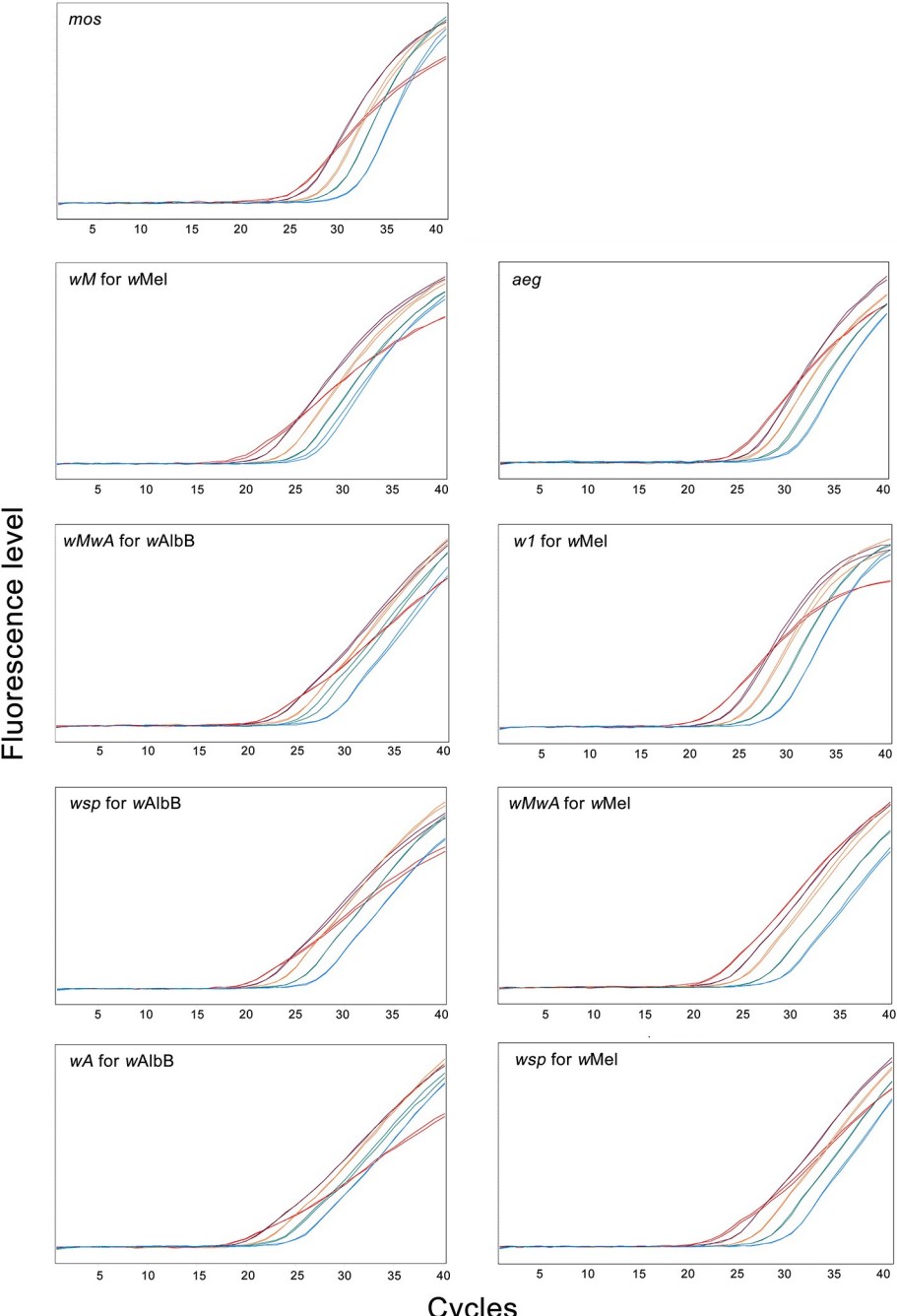

**Fig 3. Variation in the shape of the PCR amplification curves.** The curves from left to right represent amplification curves of 1/10, 1/30, 1/90, 1/270 and 1/810 DNA dilution from initial extraction in 250 μL 5% Chelex® 100 Resin. The primers are defined in Table 2.

## Conclusions

Chelex® DNA extraction and real-time PCR provide an easy and economical approach for detecting both currently-released *Wolbachia* (*w*Mel and *w*AlbB) infections in *Aedes aegypti*, while other options like multiplex probe assays and the use of DNA extraction kits are likely to cost more. Here, we designed a new primer pair, *wMwA*, which not only identifies *w*Mel and

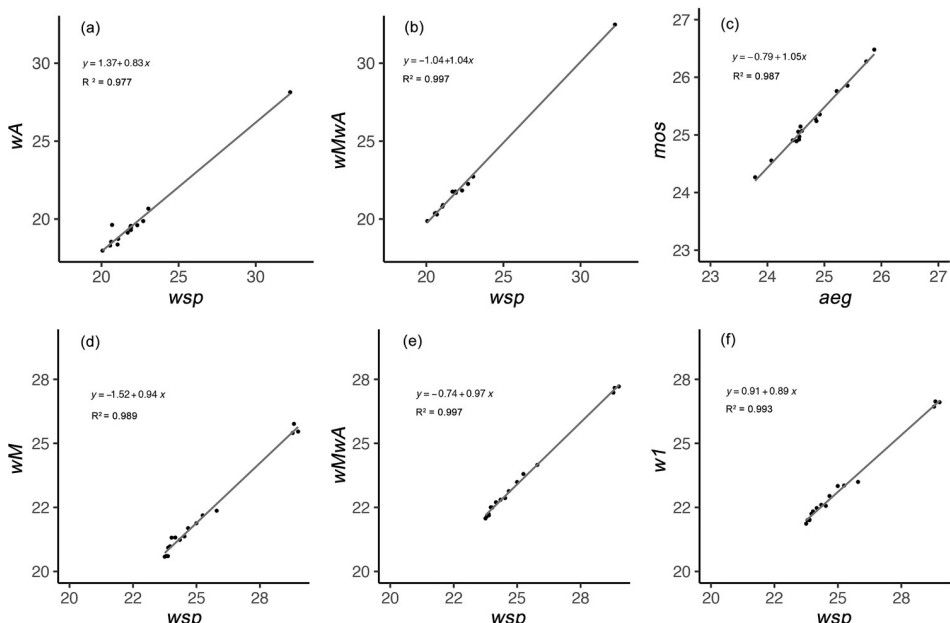

**Fig 4. Variation in C_q values when using different *Wolbachia* primers for the same samples.** Correlation of C_q values between (a) *wA* and *wsp* primers in *Wolbachia w*AlbB screening; (b) *wMwA* and *wsp* primers in *Wolbachia w*AlbB screening; (c) *mos* and *aeg* primers in *Aedes aegypti* screening; (d) *wM* and *wsp* primers in *Wolbachia w*Mel screening; (e) *wMwA* and *wsp* primers in *Wolbachia w*Mel screening; (f) *w1* and *wsp* primers in *Wolbachia w*Mel screening.

*w*AlbB at the same time, but is also correlated with density estimates based on a universal *Wolbachia* primer *wsp*. We demonstrated this new primer pair has the potential to be accurate and efficient for large-scale *Wolbachia* detection and relatively density estimates, especially for use in field collected *Ae. aegypti*.

# Supporting information

**S1 Table. Real-time PCR reagents and volume in 384-well plates with white wells.**
(DOCX)

**S2 Table. Primer efficiency when sample DNA was first diluted six times.**
(DOCX)

**S3 Table. Primer efficiency when sample DNA was extracted using different methods.**
(DOCX)

**S1 Fig. Primer efficiency when sample DNA was first diluted six times.** DNA is extracted in 250 μL 5% Chelex® 100 Resin and then diluted six times before making a three-fold dilution series. Outliers are marked with red colour and are excluded from the efficiency curve. The primer names are defined in S3 Table.
(PNG)

**S2 Fig. Variation in the shape of the PCR amplification curves when sample DNA was first diluted six times.** The curves from left to right represent amplification curves of 1/6, 1/18, 1/54, 1/162 and 1/486 DNA dilution from 250 μL 5% Chelex® 100 Resin. The primers are defined in S2 Table.
(PNG)

## Acknowledgments

We thank Perran A. Ross and Jason Axford for providing the mosquito samples. We also thank the support of the Jasper Loftus-Hills award, offered by the Faculty of Science, the University of Melbourne.

## Author Contributions

**Conceptualization:** Meng-Jia Lau, Nancy M. Endersby-Harshman.

**Formal analysis:** Meng-Jia Lau.

**Funding acquisition:** Ary A. Hoffmann.

**Investigation:** Meng-Jia Lau.

**Methodology:** Meng-Jia Lau, Nancy M. Endersby-Harshman.

**Supervision:** Ary A. Hoffmann, Nancy M. Endersby-Harshman.

**Visualization:** Meng-Jia Lau.

**Writing – original draft:** Meng-Jia Lau.

**Writing – review & editing:** Ary A. Hoffmann, Nancy M. Endersby-Harshman.

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
