## [Decision Letter · Decision Letter 0]

22 Jul 2021

PONE-D-21-20288

A diagnostic primer pair to distinguish between wMel and wAlbB Wolbachia infection

PLOS ONE

Dear Dr. Lau,

Thank you for submitting your manuscript to PLOS ONE. After careful consideration, we feel that it has merit but does not fully meet PLOS ONE’s publication criteria as it currently stands. Therefore, we invite you to submit a revised version of the manuscript that addresses the points raised during the review process.

Please try to carefully study and respond all the queries raised by both reviewers before you attempt to return your revised manuscript for further revision. There are quite important comments that need to be addressed.

Please submit your revised manuscript by  August 30th. If you will need more time than this to complete your revisions, please reply to this message or contact the journal office at plosone@plos.org. Please include the following items when submitting your revised manuscript:

We look forward to receiving your revised manuscript.

Kind regards,

Luciano Andrade Moreira, PhD

Academic Editor

PLOS ONE

Journal Requirements:

Reviewers' comments:

Reviewer's Responses to Questions

**Comments to the Author**

1. Is the manuscript technically sound, and do the data support the conclusions?

Reviewer #1: Yes

Reviewer #2: Partly

2. Has the statistical analysis been performed appropriately and rigorously? 

Reviewer #1: N/A

Reviewer #2: No

3. Have the authors made all data underlying the findings in their manuscript fully available?

Reviewer #1: Yes

Reviewer #2: Yes

4. Is the manuscript presented in an intelligible fashion and written in standard English?

Reviewer #1: Yes

Reviewer #2: Yes

5. Review Comments to the Author

Reviewer #1: In the manuscript entitled: “A diagnostic primer pair to distinguish between wMel and wAlbB Wolbachia infection”, Lau and colleague’s objective is to improve the real-time PCR diagnostic capabilities of Wolbachia, particularly the wMel and wAlbB strains, currently deployed in field settings around the globe. Based on a gene shared between the two strains (WD_RS06155), and considering the small two-base pair difference between wMel and wAlbB, the authors developed a new primer pair named wMA. Given the data provided, wMA was capable of differentiating between both bacterial strains, while displaying high efficiency and specificity parameters, particularly when compared to commonly primer sets currently employed by the community. I appreciate the concern raised by the authors over the feasibility of this new primer set as a candidate to perform relative measurements of Wolbachia density. Overall, the manuscript is well written, and presented in a clear/easy to understand way. For the most part, experiments are straightforward and well described, with a solid amount of data to support their conclusions. I have listed below, a few suggestions that I believe could improve the quality of the manuscript, prior to acceptance by the editor/Journal.

Comments are displayed in order of appearance:

Line (L) 74: Please provide in this section, more information regarding the gene. If NCBI's designation is correct, the gene is a DNA-directed RNA polymerase subunit beta/beta, with an old locus tag of WD_0024.

L. 84: Small typo right before “[13]”.

L. 88: Please specify here if only females were used for the assays, or if it also included males.

L. 91: Can the authors comment if any other extraction method was used, and if so, how the primers performed on it? For instance, the squash buffer method is commonly employed in certain laboratories as a fast method to detect and quantify Wolbachia.

L. 102: Please indicate the final volume used per reaction mix, as well as the final concentration of each primer used. I also encourage the authors to indicate the degree of transparency of the plasticware used, e.g., white or clear, since different plastics exhibit substantial differences in fluorescence reflection and sensitivity.

L. 107: I encourage the authors to change all designations of “Cp” on the manuscript, to “Cq”. Reasoning for this can be found in the quote below from the widely referenced manuscript for real-time PCR assays: DOI: 10.1373/clinchem.2008.112797

"The nomenclature describing the fractional PCR cycle used for quantification is inconsistent, with threshold cycle (Ct), crossing point(Cp), and take-off point (TOP) currently used in the literature. These terms all refer to the same value from the real-time instrument and were coined by competing manufacturers of real-time instruments for reasons of product differentiation, not scientific accuracy or clarity. We propose the use of quantification cycle (Cq), according to the RDML (Real-Time PCR Data Markup Language) data standard (http:// www.rdml.org)"

L. 112 – Table 1: Please include on this table, the database accession number for each gene of interest, and the amplicon size for each target gene.

L. 122: given the nature of the study in designing a diagnostic primer set, the authors then should include information on how often the assay returns a positive result when a target is present and how often it is negative in the absence of the target. A common requirement of diagnostic assays.

L. 140: This should not be a big issue, given the results here presented, but I encourage the authors to review their lower-end efficiency value of 85% and the citation used to back up their affirmation. Many distinct publications and guidelines for qPCR experiments encourage a range of 90%-110% as acceptable. One example of such literature can be found at DOI: 10.1016/j.tibtech.2018.12.002

L. 156 – Fig. 3: Could each curve be represented using a different color, as to facilitate the reader's visualization of the data?

Reviewer #2: This manuscript describes design and testing of a qPCR primer annealing to D_RS06155 DNA (directed RNA polymerase subunit beta/beta' gene) of Wolbachia. This primer allows to distinguish between wMel and wAlbB in Aedes aegyptii released to control vector-borne diseases. This primer is also compared to other previously published primers.

The primer's name, wMA, resembles designation of a Wolbachia strain (small "w" and Capital letter following). For clarity, shouldn't it be called something else?

This primers Blast perfectly to wMel (but also to wAu, wYak, wTei, wSan). As wAu has also been transinfected to A. aegyptii, this may limit the utility of this primer for field screening of Aedes. Also, sequences annotated as “Wolbachia endosymbiont of Aedes aegyptii” come up in a blast search quite a lot (eg. CP072672.1), which might be confusing, as these Wolbachia likely do not exist (see Ross et al. 2020). If I was designing the primer, I would avoid the possibility of possibly confusing contaminants. The quick homology searches described above seem important for the usage of the primer, but were not performed by authors and included in the text.

Lines 142-146 Shouldn’t samples with contaminants/inhibitors be removed from calibration curves and analysis? How do melting curves for 6x diluted Chelex preps look like?

Lines 160-168 – is it possible that some primer-targeted sequences have multiple copies within the two genomes? Even if they are all 1-copy sequences this is still expected, as Cp will depend on primer sequence/annealing and amplified fragment sequence.

172-173 – Different number of Wolbachia inside mosquito is to be expected, as this is what the qPCR assay is supposed to test for. The differences within Wolbachia genomes are the issue here. Is WD_RS06155 DNA a multi-copy gene in any of the strains? And, in relation to this also lines 174-175 – can you compare the wMel and wAlbB densities using this primer pairs then?

Finally, the figures presented here have very low resolution.

6. PLOS authors have the option to publish the peer review history of their article (what does this mean?). If published, this will include your full peer review and any attached files.

Reviewer #1: **Yes: **Heverton Leandro Carneiro Dutra

Reviewer #2: No

---

## [Author Response · Author response to Decision Letter 0]

21 Aug 2021

Dear reviewer/editor,

Thank you for giving us the opportunity to submit a revised draft of my manuscript titled “A diagnostic primer pair to distinguish between wMel and wAlbB Wolbachia infections”. We appreciate the time and effort that you and the reviewers have dedicated to providing valuable feedbacks on my manuscript. 

We have now addressed all the comments from the two reviewers in the newly uploaded manuscript. We believe our research has significant implications and look forward to hearing from you regarding our submission and to respond to any further questions and comments you may have.

Regards,

Meng-Jia Lau, Ary A. Hoffmann, Nancy M. Endersby-Harshman

---

## [Decision Letter · Decision Letter 1]

10 Sep 2021

A diagnostic primer pair to distinguish between wMel and wAlbB Wolbachia infections

PONE-D-21-20288R1

Dear Dr. Lau,

We’re pleased to inform you that your manuscript has been judged scientifically suitable for publication and will be formally accepted for publication once it meets all outstanding technical requirements.

Kind regards,

Luciano Andrade Moreira, PhD

Academic Editor

PLOS ONE

Additional Editor Comments (optional):

Reviewers' comments:

Reviewer's Responses to Questions

**Comments to the Author**

1. If the authors have adequately addressed your comments raised in a previous round of review and you feel that this manuscript is now acceptable for publication, you may indicate that here to bypass the “Comments to the Author” section, enter your conflict of interest statement in the “Confidential to Editor” section, and submit your "Accept" recommendation.

Reviewer #1: All comments have been addressed

Reviewer #3: All comments have been addressed

2. Is the manuscript technically sound, and do the data support the conclusions?

Reviewer #1: Yes

Reviewer #3: Yes

3. Has the statistical analysis been performed appropriately and rigorously? 

Reviewer #1: Yes

Reviewer #3: Yes

4. Have the authors made all data underlying the findings in their manuscript fully available?

Reviewer #1: Yes

Reviewer #3: Yes

5. Is the manuscript presented in an intelligible fashion and written in standard English?

Reviewer #1: Yes

Reviewer #3: Yes

6. Review Comments to the Author

Reviewer #1: (No Response)

Reviewer #3: In this manuscript the author developed a diagnostic primer pair, wMwA, that can detect and distinguish between the wMel and wAlbB infections in Aedes aegypti. Using Chelex DNA extraction and real-time PCR, the authors provide an easy and economical detecting both currently-released Wolbachia (wMel and wAlbB) infections in Aedes aegypti. As far as the data provided in the manuscript is concerned, the authors provided a robust essay where the primer pair, wMwA, was capable of differentiating between both bacterial strains. In general the experiments are well-controlled and the main conclusions are supported by the data.

Previously the reviewers 1 and 2 presented some suggestions to the manuscript, manly the primers designation and the figures resolution. In my opinion, I observed that the issues raised were addressed in this new version. Concerning the minor comments, the authors fully addressed all points raised by the reviewers.

In conclusion I recommend the publication of the presented review article

7. PLOS authors have the option to publish the peer review history of their article (what does this mean?). If published, this will include your full peer review and any attached files.

Reviewer #1: **Yes: **Heverton Leandro Carneiro Dutra

Reviewer #3: **Yes: **Alvaro Gil Araujo Ferreira

---

## [Editor Report · Acceptance letter]

16 Sep 2021

PONE-D-21-20288R1 

A diagnostic primer pair to distinguish between *w*Mel and *w*AlbB *Wolbachia* infections 

Dear Dr. Lau:

I'm pleased to inform you that your manuscript has been deemed suitable for publication in PLOS ONE. Congratulations! Your manuscript is now with our production department. 

Kind regards, 

on behalf of

Dr. Luciano Andrade Moreira 

Academic Editor

PLOS ONE